# Metabolomic Profiling and Biological Investigation of the Marine Sponge-Derived Fungus *Aspergillus* sp. SYPUF29 in Response to NO Condition

**DOI:** 10.3390/jof10090636

**Published:** 2024-09-05

**Authors:** Jiao Xiao, Xiuping Lin, Yanqiu Yang, Yingshu Yu, Yinyin Li, Mengjie Xu, Yonghong Liu

**Affiliations:** 1Wuya College of Innovation, Shenyang Pharmaceutical University, Shenyang 110016, China; yuyingshu98@163.com (Y.Y.); liyinyin1011@163.com (Y.L.); 2Guangdong Key Laboratory of Marine Materia Medica, South China Sea Institute of Oceanology, Chinese Academy of Sciences, Guangzhou 510301, China; xiupinglin@scsio.ac.cn; 3College of Information Science and Engineering, Northeastern University, Shenyang 110819, China; yangyqneu@163.com; 4Department of Biological Sciences, Xinzhou Normal University, Xinzhou 034000, China; xumj1019@126.com

**Keywords:** marine-derived fungi, marine natural product, neuroinflammation, alkaloids, UHPLC-HRMS

## Abstract

Marine-derived fungi are assuming an increasingly central role in the search for natural leading compounds with unique chemical structures and diverse pharmacological properties. However, some gene clusters are not expressed under laboratory conditions. In this study, we have found that a marine-derived fungus *Aspergillus* sp. SYPUF29 would survive well by adding an exogenous nitric oxide donor (sodium nitroprusside, SNP) and nitric oxide synthetase inhibitor (L-NG-nitroarginine methyl ester, L-NAME) in culture conditions. Moreover, using the LC-MS/MS, we initially assessed and characterized the difference in metabolites of *Aspergillus* sp. SYPUF29 with or without an additional source of nitrogen. We have found that the metabolic pathway of Arginine and proline metabolism pathways was highly enriched, which was conducive to the accumulation of alkaloids and nitrogen-containing compounds after adding an additional source of nitrogen in the cultivated condition. Additionally, the in vitro anti-neuroinflammatory study showed that the extracts after SNP and L-NAME were administrated can potently inhibit LPS-induced NO-releasing of BV2 cells with lower IC_50_ value than without nitric oxide. Further Western blotting assays have demonstrated that the mechanism of these extracts was associated with the TLR4 signaling pathway. Additionally, the chemical investigation was conducted and led to nine compounds (**SF1**–**SF9**) from AS1; and six of them belonged to alkaloids and nitrogen-containing compounds (**SF1**–**SF6**), of which **SF1**, **SF2**, and **SF8** exhibited stronger activities than the positive control, and showed potential to develop the inhibitors of neuroinflammation.

## 1. Introduction

Neuroinflammation refers to a complex process involving the activation of the brain’s innate immune system in response to an inflammatory challenge [1]. Despite the low-to-moderate grade of neuroinflammation being beneficial for repairing injured tissues and maintaining brain homeostasis [2], cellular and tissue damage could be exacerbated by excessive neuroinflammation and ultimately lead to neurodegenerative diseases [3]. Microglia, the primary immune cells of the central nervous system, are the brain’s first line of defense against injury, infection, and diseases [4]. When microglial cells respond to abnormal stimulation, high levels of pro-inflammatory cytokines including interleukin-1 beta (IL-1*β*), IL-6, tumor necrosis factor-alpha (TNF-*α*), nitric oxide (NO), etc., are released. In turn, the redundant pro-inflammatory mediators lead to further microglial activation, and cellular and brain-tissue damage [5]. Moreover, it is well known that there is increased expression of inducible NO synthase (iNOS) in the brain of a patient with Alzheimer’s disease [6]. Much research has illustrated the relationship of iNOS expression and NO generation with neuroinflammation [7]. Hence, inhibiting the excessive expressions of NO released by uncontrolled microglia is considered to be an effective therapeutic target for treating neurodegenerative diseases [8].

Natural products play a major role in the discovery of lead compounds for developing more-potent and safer drugs to treat human diseases [9,10]. The marine environment covers almost 70% of the Earth’s surface, and represents an important source of unique chemical structures and diverse pharmacological properties [11]. Considerable interest in the screening of effective marine natural products mainly focused on marine macro-organisms, in which the enormous diversity and abundance of sponges mean that sponge-derived fungi were envisaged as hot spots, due to the special environment where they lived being able to active some silent genes and induce specific metabolic pathways. The genus *Aspergillus*, especially, has proven to be an efficient producer of various bioactive metabolites that offer promise for new drug discovery [12]. As part of our continuing search for natural potential neuroinflammatory inhibitors [13,14,15,16], the fungus *Aspergillus* sp. SYPUF29 obtained from a sponge *Callyspongia* sp. attracted our attention, due to its resistance and survival rates under exogenous NO conditions. This synthase works with respect to mammalian cells; however, SNP and L-NAME are two stable compounds, and there are some articles which have proved that SNP and L-NAME, as the exogenous nitric oxide added in the culture in vitro, rather than working as a synthetase in vivo, could force fungi to alter this environment [17,18,19,20].

Therefore, this study aimed to screen the differential metabolites when *Aspergillus* sp. SYPUF29 is exposed to the cultivated condition containing additional sources of nitrogen, analyzing the metabolic pathway, and assessing the anti-neuroinflammation effects. This study established an in vitro model for fungi to alter the unique environment associated with the progression of neuroinflammation-related diseases, and to further generate and yield a series of secondary metabolites, which could be developed as important natural agents for drug discovery.

## 2. Materials and Methods

### 2.1. Preparation of Samples

Fungal material and fermentation: the marine-derived fungus *Aspergillus* sp. SYPUF29 was isolated from a sponge collected from the Guangdong Province in China. The fungal strain was maintained on potato dextrose agar (PDA) slants. The species was identified by morphology and by analysis of the ITS regions of its rDNA, whose sequence data have been deposited in GenBank (accession number PP15897). The fermentation medium of the control group contained malt extract 15 g/L, and artificial sea salt 10 g/L, H_2_O 1 L, and was adjusted to pH 7.5, while the treatment group contained SNP (0.1~1.0 mM) or L-NAME (0.1~1.0 mM), additionally. The mycelia were aseptically transferred to 500 mL Erlenmeyer flasks containing 250 mL of the MB liquid medium sterilized at 120 °C for 30 min. The flasks were incubated at 26 °C for 7 days on a rotating shaker (180 rpm). After that, the mixture was filtered through the filter paper, followed by being extracted three times with EtOAc, and was then concentrated under reduced pressure to afford an EtOAc extract.

Chemical profile: using the same condition (AS1), a large-scale fermentation (25 L) of the fungal strain *Aspergillus* sp. SYPUF29 was incubated. It was then extracted with EtOAc three times, and the EtOAc extracts were loaded on silica gel column chromatography, eluted with CH_2_Cl_2_/MeOH in gradient eluent (100:0–0:100 *v*/*v*), obtaining seven fractions Fr.1–Fr.7. Fr.2 was further portioned via an ODS column washed with MeOH-H_2_O ranging from 2:8 to 9:1 to produce 40%, and 80% elution. From the 80% elution, compound **SF9** (2.3 mg) was separated, while compounds **SF6** (5.0 mg) and **SF7** (2.0 mg) were obtained from 40% elution by reversed-phase HPLC (15% CH_3_CH_2_CN/H_2_O, 3 mL/min). Fr.3 was purified successively on the ODS column (MeOH-H_2_O, 5: 5 to 9: 1); among those fractions, 50% and 70% MeOH eluates were further chromatographed by preparative HPLC to produce compounds **SF1** (20.0 mg), and **SF2** (5.0 mg). Compound **SF8** (2.2 mg) was purified from Fr.4 via eluted HPLC. Fr.5–50% fraction was subjected to HPLC (33% CH_3_OH-H_2_O) on an RP18 column to afford compound **SF4** (10.0 mg), while compound **SF5** (2.0 mg) was produced by HPLC (72% CH_3_OH-H_2_O) from Fr.5–80% fraction. Finally, Fr.6 was isolated using HPLC with different concentrations of MeOH-H_2_O to yield compound **SF3** (30.0 mg).

Samples for untargeted mass spectrometry-based metabolomic analysis: the EtOAc extract (100 mg) was individually grounded with liquid nitrogen and the homogenate was resuspended with prechilled 80% methanol and 0.1% formic acid by well vortex. The samples were incubated on ice for 5 min and were then centrifuged at 15,000× *g*, 4 °C, for 20 min. Some of the supernatant was diluted to a final concentration containing 53% methanol by LC-MS grade water. The samples were subsequently transferred to a fresh Eppendorf tube and were then centrifuged at 15,000× *g*, 4 °C, for 20 min. Finally, the supernatant was injected into the LC-MS/MS system for analysis [21].

### 2.2. UHPLC-MS/MS Analysis

The Vanquish UHPLC system (Thermo Fisher, Frankfurt am Main, Germany) was coupled with an Orbitrap Q Exactive^TM^ HF mass spectrometer (Thermo Fisher, Germany). A Hypesil Goldcolumn (100 × 2.1 mm, 1.9 μm) was used. The mobile phases consisted of eluent A (0.1% FA in Water) and B (Methanol) at a flow rate of 0.2 mL/min for the positive polarity mode, while for the negative polarity mode, the eluent A was 5 mM ammonium acetate (pH 9.0) and eluent B was Methanol. The adopted gradient program was as follows: 2% B, 1.5 min; 2–100% B, 12.0 min; 100% B, 14.0 min; 100–2% B, 14.1 min; 2% B, 17 min.

The Q Exactive^TM^ HF mass spectrometer was operated both in positive and negative models to perform full scan monitoring on the range of *m/z* 100–1500. The other operating parameters were set as follows: spray voltage of 3.2 kV; capillary temperature of 320 °C, sheath gas flow rate of 40 arb; and aux gas flow rate of 10 arb.

### 2.3. Data Processing and Metabolite Identification

The raw LC-QTOF-MS data were analyzed using Compound Discoverer 3.1 (CD3.1, Thermo Fisher). The ions that showed a retention time tolerance of 0.2 min and an actual mass tolerance of 5 ppm, in different samples, were considered as the same ion (the other parameters including signal intensity tolerance, 30%; signal/noise ratio, 3; and minimum intensity 100,000). Then, peak intensities were normalized, to further predict the molecular formula, such as additive ions, molecular ion peaks and fragment ions, etc. All those peaks were matched and analyzed using the mzCloud (https://www.mzcloud.org/, accessed on 16 February 2022), mzVault, and MassList databases. Based on the exact masses, fragmentation pathways, retention behaviors, and related botanical biogenesis, these compounds were identified. The resulting data were analyzed using the statistical software R (R vR-3.4.3), Python (Python v2.7.6), and CentOS (CentOS release v6.6).

### 2.4. Data Analysis of Non-Targeted Metabolomics

The resulting data were subjected to multivariate analysis, using the principal component analysis (PCA) and partial least-squares discriminant analysis (PLS-DA) method, performed at metaX (v1.4.2). In the PLS-DA model, the R^2^ (goodness-of-fit variable) and Q^2^ (predictive ability variable) values were measured to evaluate the predictive ability of each model, and were not overfitted.

The statistical significance (*p*-value) was calculated using univariate analysis (*t*-test). The metabolites with VIP > 1 and *p*-value < 0.05 and fold change ≥ 2 or FC ≤ 0.5 were considered to be differential metabolites.

The functions of these metabolites and metabolic pathways were studied using the KEGG database (https://www.genome.jp/kegg/pathway.html accessed on 28 July 2024).

### 2.5. Determination of Cell Viability

BV2 microglia cell viability was evaluated by the MTT reduction assay represented in a previous work [13]. BV2 cells were seeded in the 96-well plates at a concentration of 2 × 10^5^ cells/mL, for 12 h. After pretreating with LPS and tested samples for 24 h, the cultured supernatant was collected and added to 0.25 mg/mL of MTT, and incubated for 3 h, at 37 °C. Then, the absorbance of each well was recorded on a plate reader (Bio-Tek, Winooski, VT, USA) at 490 nm.

### 2.6. NO Production Bioassay

The Griess reaction was used to measure the production of NO in LPS-induced BV2 cells. BV2 cells were plated on 96-well microtiter plates and treated with test extracts (dissolved in DMSO at the concentration of 1, 10, and 30 μg/mL), test compounds (dissolved in DMSO at the concentration of 1.25, 2.5, 5, 10, 20, and 40 μmol) in the presence of LPS (1 μg/mL) for 24 h. Then, the Griess reagent was mixed with isopycnic supernatants (50 μL) at room temperature for 15 min. After that, the absorbance was measured at 540 nm.

### 2.7. ELISA Assessment of TNF-α, IL-6, and IL-1β in Mouse Serum

The levels of IL-6 and TNF-α in the supernatants were measured using the commercial ELISA kits (Excellbio, Shanghai, China) as described in the manufacturer’s protocol. In brief, after the administration of three extracts and LPS stimulation, BV2 samples were homogenized in PBS. The mixture was centrifuged at 2000 rpm for 15 min after sonication at 4 °C. The concentrations of IL-6, IL-1β, and TNF-α were measured by ELISA kits (Excellbio, Shanghai, China). The absorbance was measured by a microplate reader (Thermo Scientific, Walthman, MA, USA).

### 2.8. RNA Isolation and Real-Time Quantitative Polymerase Chain Reaction (RT-PCR)

RT-PCR was carried out as previously described [22]. Briefly, total RNA was isolated from BV-2 microglial cells using Trizol reagents (Beyotime Biotechnology, Beijing, China) and using CFX Connect™ real-time PCR detection system (Bio-Rad, Hercules, CA, USA), GoTaq one-step real-time PCR kit, followed by quantitative RT-PCR of IL-6, IL-1β, and TNF-α. The relative mRNA expression was normalized to those of GAPDH.

### 2.9. Western Blot Analysis

Western blotting was performed to evaluate protein expression levels, as described previously [22]. Briefly, after incubation with 1, 10, and 30 μg/mL of these extracts (dissolved with DMSO), the BV2 cells were homogenized and lysed, then extracted and quantified as proteins. Each protein sample was subjected to SDS-PAGE for electrophorese, and then the proteins were transferred to a membrane, which was blocked with 5% BSA at room temperature for 1 h, and then incubated with primary antibodies overnight at 4 °C. The following day, the membranes were incubated secondary anti-bodies linked with horseradish peroxidase. Using ECL reagent (Tanon, Shanghai, China), and ImageJ software (v9) for quantitative analyses.

## 3. Results

### 3.1. Phylogenetic Analysis

Fungi are the main source of microbial natural products, with chemical diversity, high yield, and diverse pharmacological properties [23]. In the case especially of the genus *Aspergillus*, the most common and ubiquitously distributed fungi on earth have received our attention. During our ongoing research aimed at the discovery of natural inhibitors of neuroinflammation, it was found that an extract from a strain *Aspergillus* sp. SYPUF29 exerted significant activities on overactivated microglial cells under additional NO conditions. The strain *Aspergillus* sp. SYPUF29 was isolated from a marine sponge collected from the Guangdong Province in China. The ITS gene sequence of SYPUF29 was PCR-amplified, sequenced, and compared to GenBank, which indicated that the strain SYPUF29 was closely associated with the genus *Aspergillus*. Phylogenetic analysis (Figure 1A) based on the ITS gene sequence (Gene Bank ID: PP15897, described in the Appendix A) revealed that strain SYPUF29 formed a distinct phylogenetic cluster with *A. aculeatus* in the phylogenetic tree.

### 3.2. Fermentation Conditions for Aspergillus *sp.* SYPUF29 to Produce the Differential Metabolites under an Additional Source of Nitrogen

To investigate whether *Aspergillus* sp. SYPUF29 could survive well and produce chemo-preventive reagents against NO, we determined the HPLC profiles of this strain in MB medium with a concentration gradient of NO donor (Sodium Nitroprusside (SNP)) and NO synthetase inhibitor (L-NAME) after 7 days. Compared with the control group (without NO), there were significant differences in both characteristic ingredients and content of the common components in the L-NAME and SNP groups (Figure 1B).

Furthermore, we also evaluated the anti-neuroinflammatory activities of extracts after SNP and L-NAME in different concentrations (0.1, 0.3, 1.0 mM) in LPS-induced BV2 microglial cells by monitoring NO production. To avoid the effects of reduced cytotoxicities on NO release, MTT assays were measured to evaluate the cell viability of these samples on BV2 cells [15]. As shown in Figure 1C, the crude extracts of *Aspergillus* sp. SYPUF29 under 0.1 mM of SNP (AS1) and 0.3 mM of L-NAME (AL3) significantly decreased NO accumulation, and no obvious cytotoxicities were observed at the effective concentrations.

### 3.3. Multivariate Data Analysis for Extract Samples

Based on the aforementioned experimental data, we thus questioned whether the AS1 and AL3 contain different components compared with the control group. Thus, a comparative analysis between the crude extracts of the three fermentation conditions was performed, using the principal component analysis (PCA) and orthogonal partial least-squares discriminant analysis (PLS-DA), with data from three groups.

Initially, PCA was performed on each study group separately, aiming to evaluate the quality and homogeneity of the data. The sample dispersions with higher similarities are closer together, while samples with bigger differences are farther apart [24], indicating high variations between the three groups. PCA score plots indicated that the metabolic profiles of three groups, both in positive (Figure 2A) and negative (Figure 2B) ion mode, differed markedly.

To further investigate the degree of similarity and differences between the two groups, a loading plot of PLS-DA was carried out. It had shown a great goodness-of-fit (*R*^2^) and high predictability (Q^2^) (Figure 2C,D), with 0.84 and −1.17 in positive mode and 0.70 and −1.32 in negative mode for the samples of AS1 vs. AC. Similarly, both the high R^2^ (goodness-of-fit) of 0.89 and Q^2^ (goodness-of-prediction) of −1.12 indicate the good fitness and predictive power of the AL3 vs. AC. All permuted R^2^ and Q^2^ values on the left were lower than the original point on the right, and the Q^2^ regression line in red had a negative intercept. Therefore, the validity of the established PLS-DA models was strongly confirmed [25].

### 3.4. Effect of SNP and L-NAME on the Metabolites in the Aspergillus sp. SYPUF29

To investigate the effect of SNP and L-NAME on the metabolites of *Aspergillus* sp. SYPUF29, differential metabolites were identified. In total, 946, 520 compounds were identified or tentatively characterized in the positive and negative models, respectively. Among them, 270 (103 in the positive model, 167 in the negative model) and 93 (38 in the positive model, 55 in the negative model) metabolite ion features differed from the culture systems in the SNP (AS1) and L-NAME (AL3) system mode, respectively.

To find out the different metabolites between the three extracts, the metabolites were chosen by the fold change (FC > 1.2 or FC < 0.833) with significant differences (*p* < 0.05), and, further transforming the fold changes (FCs) to log_2_FC, the up-regulated metabolites have a positive number of log_2_FCs. Similarly, the negative number of log_2_FCs indicated down-regulated metabolites. As shown in Figure 3A–D, the volcanic map analysis based on the different components in three extracts demonstrated that the majority of the differential metabolites were significantly upregulated after being SNP-stimulated. In the positive model, although all two extracts of AS1 and AL3 yield a set of 9 shared metabolites, the AS1 vs. AC exhibited the largest number of differential metabolites (158 in number), while that number in the AL3 vs. AC set was only 46 (Figure 3E). Similarly, it is also noteworthy that AS1 showed the presence of a higher number of total metabolites in the negative model (Figure 3F).

Taken together, it was demonstrated that strain *Aspergillus* sp. SYPUF29 can generate different metabolites in response to the exogenous NO, although the three extracts shared few common components.

### 3.5. Difference in Metabolites between Three Extracts

We further chose the VIP values of over 1.5 to figure out the changes in components after AS1 and AL3 were stimulated. Here, we obtained 28, 20, 12, and 16 different metabolites in the AS1 vs. AC (Appendix A, Appendix A) and AL3 vs. AC (Appendix A, Appendix A) in the positive model, and AS1 vs. AC (Appendix A, Appendix A) and AL3 vs. AC (Appendix A, Appendix A) in the negative model after the screening, respectively (Table 1). Except for the groups of AL3-AC in the negative model, our results have shown that many alkaloids and nitrogen-containing compounds (except for the nucleotides, amino acids, and their derivatives) presented more abundantly in AS1 and AL3. Thus, it was proved that the diagnostic performance of metabolites was accumulated in the fermentation when the *Aspergillus* sp. SYPUF29 fungus was cultivated on the culture media containing SNP.

### 3.6. KEGG-Enriched Pathway Analysis

Based on KEGG annotation analysis, the differential metabolites were significantly enriched in many metabolic pathways, with Pantothenate and CoA biosynthesis, Taurine and hypotaurine metabolism, Glyoxylate and dicarboxylate metabolism, and Arginine and proline metabolism pathways being strengthened by adding SNP in cultured conditions (Figure 4A,B).

In terms of the AL3 vs. AC set, several differential metabolites were identified, while the metabolic pathways were not observed significantly (Figure 4C,D).

Therefore, it is implied that the accumulation of alkaloids and nitrogen-containing compounds after SNP administration is related to the highly enriched Arginine and proline metabolism pathways, which needs further study to clarify.

### 3.7. The Anti-Neuroinflammatory Effects of AS1 and AL3 in LPS-Treated BV2 Cells

The IL-1*β*, IL-6, and TNF-a can also reflect the activation level of microglia, and their expression after AS1 and AL3 were induced was further explored. Real-time PCR and ELISA assays were performed to measure pro-inflammatory cytokines changes. As shown in Figure 5A,B, compared with the blank group, higher mRNA expression and protein levels of TNF-*a*, IL-1*β*, and IL-6 were observed in both ASL and AL3.

TLR4 (Toll-like receptor 4), a primary receptor of the innate immune response, is expressed in microglia and mediates microglial activation [26]. When the microglial cells are exposed to an abnormal situation, the TLR4/MD-2 complex on the cell surface will interact with LPS, and the TLR4/MD2 complex recruits the downstream adaptors such as myeloid differentiation primary response gene 88 (MyD88) [27]. The MyD88-dependent pathway further triggers the activation and promotes the nuclear entry of nuclear factor kappa-B (NF-κB), leading to the transcription of various pro-inflammatory proteins, such as TNF-*a*, IL-1*β*, IL-6, and NO [28]. Therefore, we measured the bioactivity of AS1 and AL3 on the proteins of the TLR4 signaling pathway (TLR4, MyD88, phosphorylated IκBα, phosphorylated p65, and iNOS) in the overactivated BV2 cells by Western blotting (Figure 5C). As a result, these protein levels were enhanced in the LPS-induced cells and significantly decreased with the treatment of AS1 and AL3. In most cases, AS1 has stronger inhibitory effects than AL3, both in 10 and 30 μg/mL. Taken together, our data suggested that AS1 and AL3 had strong anti-neuroinflammatory effects in LPS-stimulated BV2 cells in the manner of the TLR4/MyD88/NF-κB/iNOS-dependent mechanism.

### 3.8. The Chemical Basis of Aspergillus *sp.* SYPUF29 after SNP Was Administrated

As the mentioned above, we further focused on clarifying the chemical basis of *Aspergillus* sp. SYPUF29 with additional SNP. Chemical investigation resulted in the isolation and structural elucidation of nine compounds (Figure 6), including six alkaloids and nitrogen-containing compounds (**SF1**, JBIR-74; **SF2**, JBIR-75; **SF3**, *N*-acetyl-*β*-oxotryptamine; **SF4**, 1*H*-indole-3-carbaldehyde; **SF5**, indole-3-carboxamide; **SF6**, *N*-(2-hydroxyphenyl) acetamide), two phenolics (**SF7**, tyrosol; **SF8**, terphenyllin), and one steroid (**SF9**, 3*β*, 15*α*-dihydroxyl-(22*E*, 24*R*)-ergosta-5, 8(14), 22-trien-7-one), and their structural elucidation (including NMR and LCMS data) are listed in Appendix A, Appendix A. We further evaluated the anti-neuroinflammatory activities of the isolated compounds. As shown in Table 2, alkaloids of **SF1**, **SF2,** and phenolics of **SF8** exhibited stronger activities than the positive control, and the alkaloidal isolate **SF3** had moderate protective effects at the test concentrations. This finding indicated that those compounds generated by *Aspergillus* sp. SYPUF29 with additional SNP, were promising natural neuroinflammatory agents.

## 4. Discussion

Screening for new drug candidates for treating health issues such as cancer, neurodegenerative pathologies, and antibiotic resistance, has received much attention. Recently, funding and exploring natural products from microbial sources, especially from the marine-derived, has attracted renewed interest. Among them, alkaloids with the backbone of several nitrogenated structures have been termed “privileged structures” concerning pharmaceutical development [29,30], with heterocyclic alkaloids, especially, showing properties such as significant anti-bacteria, anti-fungi, anti-protozoa, and so forth. Importantly, some of them inhibit depression and anxiety [31]. Notably, based on the indole alkaloids, vilazodone—“Viibryd”—has been approved by the FDA for treating depressive disorder in a clinical setting.

In humans, Nitric Oxide (NO) is a signaling molecule distributed to almost all endothelial and neuronal cells. NO plays an essential role in homeostatic functions, such as regulating immune processes involving neuromodulation and neurotransmission, the regulation of platelet aggregation, etc. [32]. Despite its homeostatic activity, NO is not always beneficial. Within the inflamed brain, over-activated microglia and astrocytes will produce large amounts of nitric oxide [33], and, in return, excessive accumulation of nitric oxide will cause neuronal toxicity and death, and eventually lead to neuroinflammation. Moreover, clinical evidence has proved that increased NO production occurs in patients with neuroinflammation- and neurodegeneration-related conditions [32].

Nitric oxide (NO) is widely conserved among organisms, including microorganisms (such as bacteria, yeasts, and fungi), and higher eukaryotes (such as plants and mammals). In mammals, NO is mainly generated by NO synthase (NOS) when conversing L-arginine to L-citrulline [34], distributed to almost all endothelial and neuronal cells. A similar process for NO generating occurs by NOS or nitrate/nitrite reductase in higher plants [35]. For fungi, although most of them are unlikely to possess ortholog genes as NOS in mammals, some indications for the occurrence of NO synthase have been proposed [36,37]. As reported earlier, it has been proved that fungi have an effective enzymatic repertoire to alleviate the reactive nitrogen species (such as peroxynitrite (ONOO−) and nitric oxide (NO)) produced by the host [37]. Previously published literature has found that the performance of the fungus *Neurospora crassau* was affected by changing the cultivation medium to nitric oxide donor (sodium nitroprusside, SNP or S-nitrosoglutathione) and NOS inhibitor (L-nitroarginine) [38]. Moreover, Wang et al. have demonstrated that fungal secondary metabolite biosynthesis would be regulated, and generated the production of photoactive perylenequinones, by SNP [17]. Therefore, based on the aforementioned reports, considering the imbalance of NO content in neuroinflammation, we proposed a strategy to force fungus to alter the condition of the overdose of NO, and then generate some chemical agents for treating neurodegeneration-related diseases.

Screening more structurally diverse alkaloids from marine-derived species is a vital strategy for discovering agents to treat neuroinflammation-related diseases. In this study, an OSMAC (One Strain Many Compounds) approach was followed, together with cocultivation for the induction of cryptic neuroinflammatory inhibitors of the fungus *Aspergillus* sp. SYPUF29. Under the additional source of nitrogen, the extracts of the strain *Aspergillus* sp. SYPUF29 showed lower IC_50_ values than without using the NO-stimulated cultivate condition. These findings suggested that nitrogen, as an additional source, allows the fungus to produce large amounts of compounds responsible for the anti-neuroinflammation activities.

In addition, using the LC-MS/MS, we initially assessed and characterized the difference in metabolites of *Aspergillus* sp. SYPUF29 with or without an additional source of nitrogen. Untargeted mass spectrometry-based metabolomic analysis showed that the metabolic pathway of Arginine and the proline metabolism pathways were highly enriched, which was conducive to the accumulation of alkaloids and nitrogen-containing compounds after SNP administration. Moreover, we also analyzed the anti-inflammation activities of those differential metabolites in Appendix A. We have found that compound **27** (Agnuside) in Appendix A could be a nutraceutical for allergic asthma [39]. Similarly, eucalyptol (**12**, in Appendix A) could reduce the expression level of inflammation cytokines, and control airway mucus hypersecretion [40]. Corticosterone (**11**, in Appendix A) has been reported as a treatment for alcoholic tissue injury at the gut–liver–brain axis, acting on alcohol-induced expression of inflammatory cytokines [41]. For compound corticosterone, isorhamnetin-3-glucoside (**4**, in Appendix A), has been regarded as the principal substance responsible for anti-inflammatory activity [42].

Furthermore, we measured the anti-neuroinflammation activities of these extracts and isolates (**SF1**–**SF9**) from AS1, in LPS-induced BV2 cells. Among them, compound **SF4** is the same one as that identified from the extract of AS1 (Appendix A, compound 18), and **SF3**–**SF5** all belong to the family of indole alkaloids derived from the tryptophan. In addition, **SF4** (indole-3-carboxaldehyde) could inhibit intestinal inflammation on the TLR4/NF-kB/p38 signaling pathway, and balance amino acid metabolism in mice with colitis [43]. 2-Acetamidophenol (**SF6**) is an aromatic compound with the potential for anti-inflammatory activity, and has been widely used in the pharmaceutical industry [44]. **SF8** (terphenyllin) could suppress the growth and metastasis of gastric cancer by inhibiting the STAT3 signaling pathway [45], also showing significant neuroprotective effects against oxidative stress in Neuro-2a Cells [46]. Taken together, our results indicated that alkaloids and nitrogen-containing compounds generated by *Aspergillus* after SNP was administered have shown potential to treat inflammation-related disease, while further in-depth biological testing needs to be carried out.

## Figures and Tables

**Figure 1 jof-10-00636-f001:**
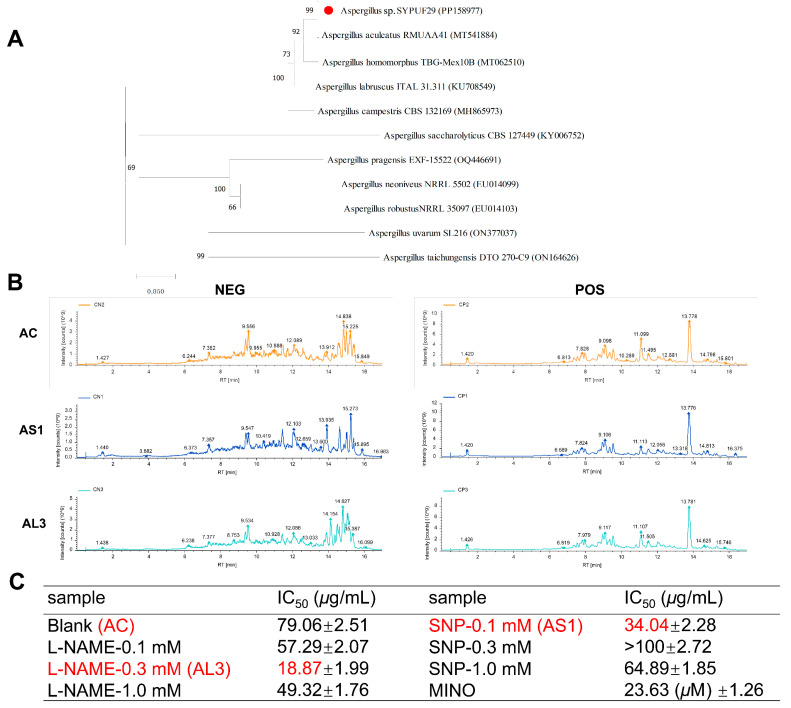
(**A**) The morphology by analysis, and neighbor-joining tree based on ITS sequences showing relationships between strain *Aspergillus* sp. SYPUF29 and closely related members of *Aspergillus* sp.; (**B**) the base peak ion (BPI) of the metabolites of *Aspergillus* sp. SYPUF29 with/without NO in the positive and negative mode; (**C**) effects of extracts after SNP and L-NAME administrated on NO production in LPS-activated BV-2 microglia cells (L-NAME-0.1/0.3/1.0 mM: the EtOAc extracts when *Aspergillus* sp. SYPUF29 was exposed to differential concentrations of L-NAME; SNP-0.1/0.3/1.0 mM: the EtOAc extracts when *Aspergillus* sp. SYPUF29 was exposed to differential concentrations of SNP; MINO: minocycline, as the positive control); IC_50_ represents the concentration of extracts that is required for 50% inhibition of NO production.

**Figure 2 jof-10-00636-f002:**
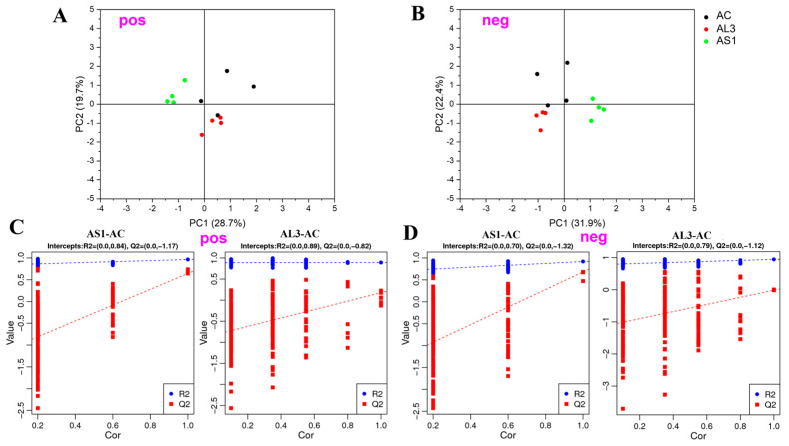
(**A**,**B**) PCA plot between the AS1, AL3, and AC in positive− and negative−ion mode. (**C**,**D**) The parameters of the PLS−DA model from AS1 vs. AC, AL3 vs. AC in positive− and negative−ion mode.

**Figure 3 jof-10-00636-f003:**
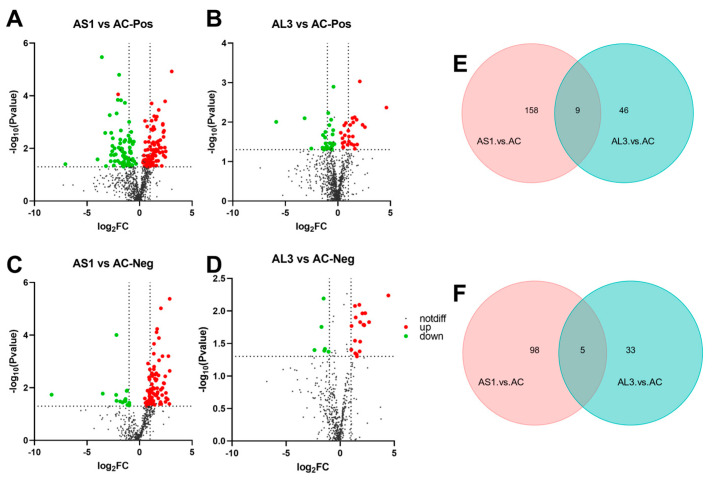
The volcano plots of positive (**A**,**C**) and negative (**B**,**D**) modes based on the non−target metabolomics in the active extracts of *Aspergillus* sp. SYPUF29. High expression is shown by the red color, and low expression is shown by the green color. (**E**,**F**) Two−way Venn diagrams showing the unique and shared metabolites in both positive and negative models for the three groups.

**Figure 4 jof-10-00636-f004:**
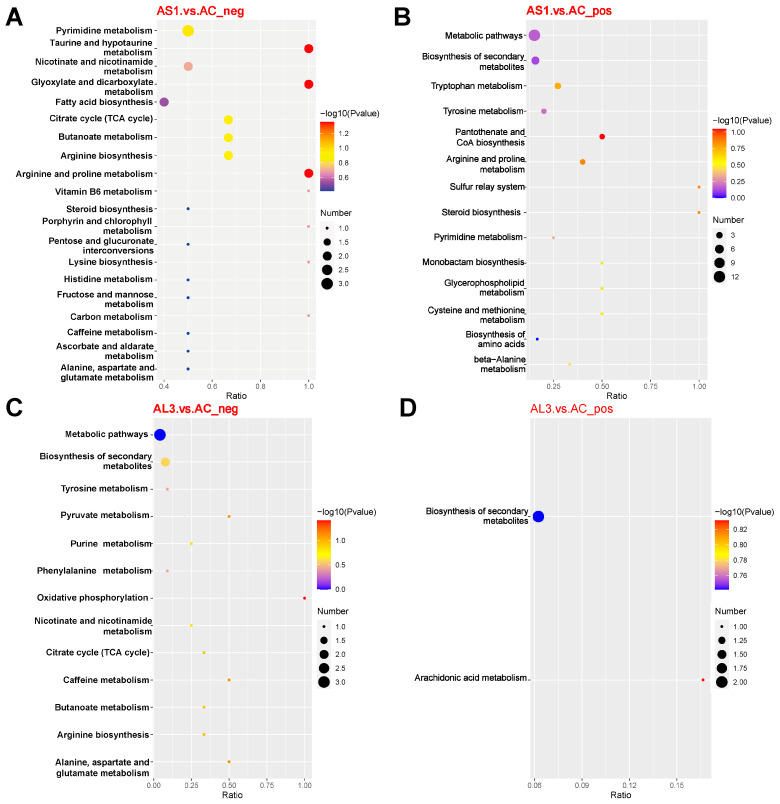
The KEGG-enriched pathways analysis of the AS1 vs. AC and the AL3 vs. AC in both positive (**B**,**D**) and negative (**A**,**C**) models.

**Figure 5 jof-10-00636-f005:**
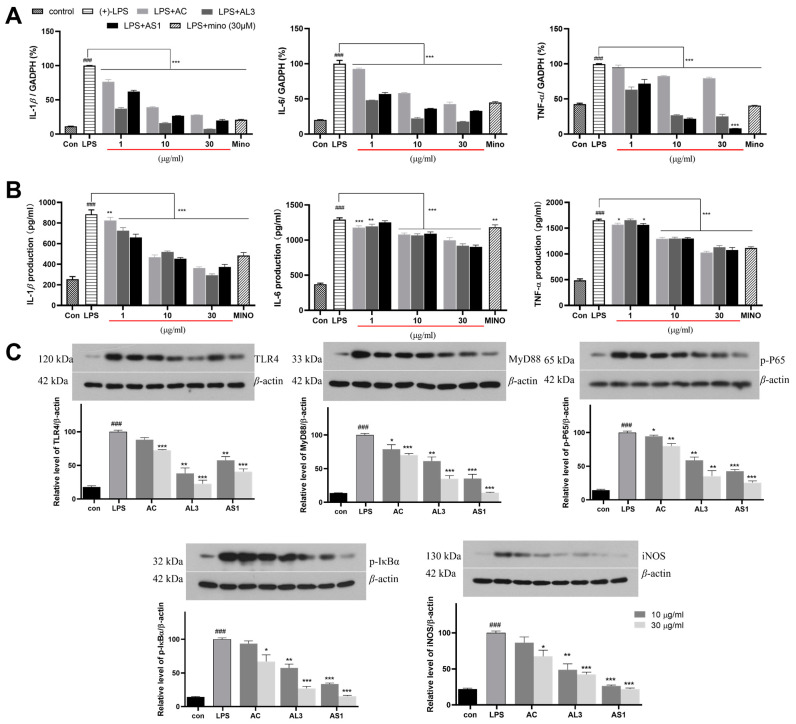
The effect of these extracts on the expressions of pro-inflammatory cytokines IL-1*β*, IL-6, and TNF-α in LPS-activated BV-2 cells using ELISA (**A**) and real-time PCR (**B**) and assay; (**C**) the protein levels of TLR4, MyD88, p-P65, p-IκB-α, and iNOS were examined by Western blot. Data are presented as mean ± SEM (n = 3). ^###^ *p* < 0.001 vs. control group; * *p* < 0.05, ** *p* < 0.01, *** *p* < 0.001 compared with the LPS group.

**Figure 6 jof-10-00636-f006:**
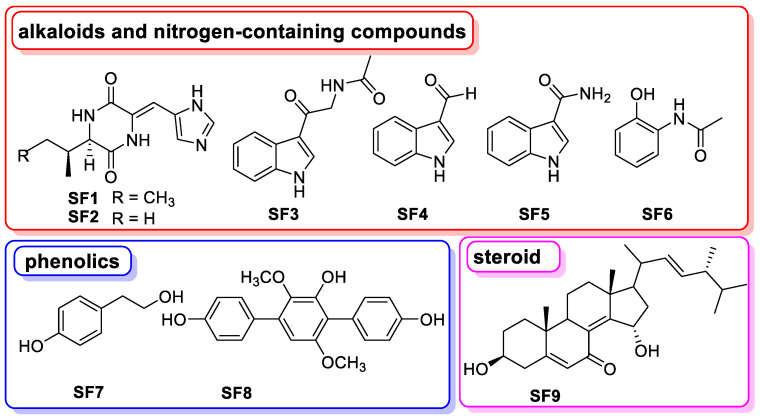
The structures of compounds isolated from *Aspergillus* sp. SYPUF29 after SNP was administrated (AS1).

**Table 1 jof-10-00636-t001:** The numbers of differential-type metabolites among the three extracts.

Types	AS1-AC-Pos	AL3-AC-Pos	AS1-AC-Neg	AL3-AC-Neg
Alkaloids and Nitrogen-containing compounds	14	9	3	2
Flavonoids	1	1	0	2
Amino acids and their derivatives	3	1	0	3
Nucleotides and their derivatives	0	1	1	0
Phenolic acids	4	2	1	1
Esters	0	0	1	0
Terpenes	0	2	0	0
Steroids	3	2	2	0
others	1	0	0	0
Phospholipids	0	0	2	3
Lipids	0	2	2	4
Coumarins	1	0	0	0
Organic acids	1	0	0	1
Total	28	20	12	16

**Table 2 jof-10-00636-t002:** Effects of pure compounds of AS1 on NO production by LPS-activated microglia cells.

Compounds	IC_50_ (μM)	Compounds	IC_50_ (μM)
**SF1**	13.19	**SF6**	>40
**SF2**	1.73	**SF7**	>40
**SF3**	38.99	**SF8**	2.78
**SF4**	>40	**SF9**	>40
**SF5**	>40	Mino	23.63

Mino: minocycline, as the positive control.

## Data Availability

The original contributions presented in the study are included in the article/Appendix A, further inquiries can be directed to the corresponding authors.

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
