# Peer review of "Metabolomic Profiling and Biological Investigation of the Marine Sponge-Derived Fungus Aspergillus sp. SYPUF29 in Response to NO Condition"

_jof, 2024, doi:10.3390/jof10090636_

Round 1

Reviewer 1 Report (New Reviewer)

The manuscript by Jiao Xiao et al. aims to demonstrate metabolomic profiling and biological activity of the fungus Aspergillus sp. SYPUF29 in response to NO exposure.  Although the manuscript contains potentially novel information, it has a number of shortcomings that preclude publication in this form. In particular:

1) FIG. 1 (C) - how can we explain the fact that when 0.3 mM NO synthetase inhibitor was added to the tested samples, a significant decrease in NO accumulation was observed, while increasing or decreasing the concentration of L-NAME reduced these values by more than half, the same is true for the experiment with the addition of sodium nitroprusside.

2) Figure 5A and 5B - correct LPS+SNP to LPS+AC1.

3) 3.8 Chemical basis of Aspergillus sp. SYPUF29 after introduction of SNPs - it is not clear from which data the detailed chemical structure of the compounds was derived, including absolute configurations of asymmetric centers. Please explain in detail.

The manuscript by Jiao Xiao et al. aims to demonstrate metabolomic profiling and biological activity of the fungus Aspergillus sp. SYPUF29 in response to NO exposure.  Although the manuscript contains potentially novel information, it has a number of shortcomings that preclude publication in this form. In particular:

1) FIG. 1 (C) - how can we explain the fact that when 0.3 mM NO synthetase inhibitor was added to the tested samples, a significant decrease in NO accumulation was observed, while increasing or decreasing the concentration of L-NAME reduced these values by more than half, the same is true for the experiment with the addition of sodium nitroprusside.

2) Figure 5A and 5B - correct LPS+SNP to LPS+AC1.

3) 3.8 Chemical basis of Aspergillus sp. SYPUF29 after introduction of SNPs - it is not clear from which data the detailed chemical structure of the compounds was derived, including absolute configurations of asymmetric centers. Please explain in detail.

Author Response

Review 1:

The manuscript by Jiao Xiao et al. aims to demonstrate metabolomic profiling and biological activity of the fungus Aspergillus sp. SYPUF29 in response to NO exposure. Although the manuscript contains potentially novel information, it has a number of shortcomings that preclude publication in this form. In particular:

1) Fig. 1 (C) - how can we explain the fact that when 0.3 mM NO synthetase inhibitor was added to the tested samples, a significant decrease in NO accumulation was observed, while increasing or decreasing the concentration of L-NAME reduced these values by more than half, the same is true for the experiment with the addition of sodium nitroprusside.

Response: Thank you for your valuable consideration. Our previous study has found that, compared to the control group, the peak areas of differential metabolites on HPLC profiles under differential concentrations of exogenous nitric oxide were distinctive, which could explain the results of anti-neuroinflammation activities.

2) Figure 5A and 5B - correct LPS+SNP to LPS+AC1.

Response: Thank you for your careful review, and we have revised it.

3) 3.8 Chemical basis of Aspergillus sp. SYPUF29 after introduction of SNPs - it is not clear from which data the detailed chemical structure of the compounds was derived, including absolute configurations of asymmetric centers. Please explain in detail.

Response: Thank you for your valuable consideration, we have added the NMR data, LCMS data, and the configurations of those compounds and their references in Figure S5-S30, Table S3-S9.

Reviewer 2 Report (New Reviewer)

This manuscript requires additional major revision.

Based on the previous list of comments, it can be seen that:

1. In section 3.1, the authors should fully describe the establishment of the phylogenetic location, namely, to which section and series of the genus Aspergillus this strain can be attributed. Which species is it closest to. Figure 1 A is hard to read.

2. The name of section 3.2 has been corrected.

3. The introduction is very poorly written. There is no clearly defined goal. At the same time, the authors describe the results and conclusions already obtained in the Introduction. I remind you that the Introduction should contain the following information: relevance, novelty and purpose of the study. I.e., the introduction should end with a text like this: "the purpose of our study is to study the effect of moonlight on the growth of telegraph wires in North Antarctica in the conditions of the polar night." All the results and conclusions are described in the relevant sections of the manuscript.

4. The remark was taken into account.

5. Identification of the compounds in UPLC MS data. 

6. The remark was taken into account.

7. The authors cited two articles [17, 18] to confirm that sodium nitroprusside (SNP) enhanced the activity of nitric oxide synthase. Moreover, [19, 20] confirmed that L-NAME is a non-selective NOS inhibitor preventing excess NO production. However, these works concern mammalian cells. Fungi, of course, are eukaryotes, but they have significant differences and it is impossible to transfer data on mammals to fungi. Thus, there is no evidence that these inductors act this way. Thus, the design of the study and the conclusions cannot be considered justified.

8. IC50 data (is it mean?) in the Figure 1c had not standard error of mean. 

9. The remark was taken into account.

10. Table S5.

11. As for the Discussion. The authors continue to discuss the Nitric oxide (NO) based on data on mammals and humans (Lines 371-380). But Fungi are not mammals! The authors continue to ignore the fact that fungi have other signaling pathways, although somewhat similar. The authors ignored the recommendation to discuss the anti-inflammatory activity of the compounds. In the discussion (lines 396-409), the results are essentially repeated and there is no analysis of the literature on already known literary data. I.e., the authors need to discuss what activity has already been shown for substances, and how this can ensure their activity described in this manuscript.

12. The remark was taken into account.

13. In the Discussion, the authors took into account the remark that the additives used were probably only an additional source of nitrogen for the biosynthesis of nitrogen-containing compounds. However, nothing has changed in setting the goal. That is, now the discussion of the results does not correspond to the goal. Or the goal does not match the results. In general, you need to revise the entire manuscript from beginning to end.

Additional comments.

14. The authors identified several compounds and described the isolation scheme. However, there is no information about how these compounds were identified (their structure was established). Are these substances new or known? 

15. The headings of some tables and figure captions are incorrect.

Author Response

Review 2:

Based on the previous list of comments, it can be seen that:

  1. In section 3.1, the authors should fully describe the establishment of the phylogenetic location, namely, to which section and series of the genus Aspergillus this strain can be attributed. Which species is it closest to. Figure 1 A is hard to read.

Response: Thans for your comment. We have revised as: The strain Aspergillus sp. SYPUF29 was isolated from a marine sponge collected from the Guangdong Province in China. The ITS gene sequence of SYPUF29 was PCR-amplified, sequenced, and compared to GenBank, which indicated that the strain SYPUF29 was closely associated with the genus Aspergillus. Phylogenetic analysis based on ITS gene sequence (Gene Bank ID: PP15897, described in the Supporting Information) revealed that strain SYPUF29 formed a distinct phylogenetic cluster with A. aculeatus in the phylogenetic tree.

  1. The name of section 3.2 has been corrected.

Response: Thank you.

  1. The introduction is very poorly written. There is no clearly defined goal. At the same time, the authors describe the results and conclusions already obtained in the Introduction. I remind you that the Introduction should contain the following information: relevance, novelty and purpose of the study. I.e., the introduction should end with a text like this: "the purpose of our study is to study the effect of moonlight on the growth of telegraph wires in North Antarctica in the conditions of the polar night." All the results and conclusions are described in the relevant sections of the manuscript.

Response: Thank you for your valuable suggestion, we have revised it in this manuscript, as follows: Therefore, this study aimed to screen the differential metabolites when Aspergillus sp. SYPUF29 is exposed to the cultivated condition containing additional sources of nitrogen, analyzes the metabolic pathway, and assesses the anti-neuroinflammation effects. This study established an in vitro model for fungi to alter the unique environment associated with the progression of neuroinflammation-related diseases, and further generate and yield a series of secondary metabolites, which could be developed as important natural agents for drug discovery.

  1. The remark was taken into account.

Response: Thank you.

  1. Identification of the compounds in UPLC MS data.

Response: Thank you for your valuable comments, although compounds SF1-SF9 were identified using NMR method (Table S3-S9), as you suggest, we also identified them by HPLC-MS, and these data were listed in Figure S5-S30, Table S1-S5.

  1. The remark was taken into account.

Response: Thank you.

  1. The authors cited two articles [17, 18] to confirm that sodium nitroprusside (SNP) enhanced the activity of nitric oxide synthase. Moreover, [19, 20] confirmed that L-NAME is a non-selective NOS inhibitor preventing excess NO production. However, these works concern mammalian cells. Fungi, of course, are eukaryotes, but they have significant differences and it is impossible to transfer data on mammals to fungi. Thus, there is no evidence that these inductors act this way. Thus, the design of the study and the conclusions cannot be considered justified.

Response: Thank you for your consideration, we highly agree with your opinion that “these synthase works concern mammalian cells.” However, SNP and L-NAME are two stable compounds, and there are some articles have proved that SNP and L-NAME as the exogenous nitric oxide added in the culture in vitro, not worked as the synthetase in vivo, could force fungi alter this environment [1–4].

  1. Wang, W.J.; Li, X.P.; Shen, W.H.; Huang, Q.Y.; Cong, R.P.; Zheng, L.P.; Wang, J.W. Nitric Oxide Mediates Red Light-Induced Perylenequinone Production in Shiraia Mycelium Culture. Bioresources and Bioprocessing 2024, 11, 2, doi:10.1186/s40643-023-00725-5.
  2. Daroodi, Z.; Taheri, P. Function of the Endophytic Fungus Acrophialophora Jodhpurensis, Methionine, and Nitric Oxide in Wheat Resistance Induction against Fusarium Graminearum via Interplay of Reactive Oxygen Species and Iron. Physiological and Molecular Plant Pathology 2023, 128, 102132, doi:10.1016/j.pmpp.2023.102132.
  3. Modolo, L.V.; Cunha, F.Q.; Braga, M.R.; Salgado, I. Nitric Oxide Synthase-Mediated Phytoalexin Accumulation in Soybean Cotyledons in Response to the Diaporthe Phaseolorumf. Sp. Meridionalis Elicitor. Plant Physiology 2002, 130, 1288–1297, doi:10.1104/pp.005850.
  4. Gong, X.; Fu, Y.; Jiang, D.; Li, G.; Yi, X.; Peng, Y. L-Arginine Is Essential for Conidiation in the Filamentous Fungus Coniothyrium Minitans. Fungal Genetics and Biology 2007, 44, 1368–1379, doi:10.1016/j.fgb.2007.07.007.
  5. IC50 data (is it mean?) in the Figure 1c had not standard error of mean.

Response: Thanks for your comments. IC50 represents the concentration of extracts that is required for 50% inhibition of NO production, which was added in the legend. And we have added the error of mean in Figure 1C.

  1. The remark was taken into account.

Response: Thank you.

  1. Table S5.

Response: the corresponding data of original Table S5 was replaced by figure S1-S4 in the most recent version of the manuscript.

  1. As for the Discussion. The authors continue to discuss the Nitric oxide (NO) based on data on mammals and humans (Lines 371-380). But Fungi are not mammals! The authors continue to ignore the fact that fungi have other signaling pathways, although somewhat similar. The authors ignored the recommendation to discuss the anti-inflammatory activity of the compounds. In the discussion (lines 396-409), the results are essentially repeated and there is no analysis of the literature on already known literary data. I.e., the authors need to discuss what activity has already been shown for substances, and how this can ensure their activity described in this manuscript.

Response: Thanks for your comments. As mentioned above, we highly agree with your idea that “Fungi are not mammals”. However, there are some articles have proved that SNP and L-NAME as the exogenous nitric oxide added in the culture in vitro, not worked as the synthetase in vivo, could force fungi alter this environment. And we have added more information to clarify this situation in discussion part, as follows: Nitric oxide (NO) is widely conserved among organisms, including microorganisms (such as bacteria, yeasts, and fungi), and higher eukaryotes (such as plants and mammals). In mammals, NO is mainly generated by NO synthase (NOS) when conversing L-arginine to L-citrulline [30], distributed to almost all endothelial and neuronal cells. A similar process for NO generating occurs by NOS or nitrate/nitrite reductase in higher plants [31]. For fungi, although most of them are unlikely to possess ortholog genes as NOS in mammals, some indications for the occurrence of NO synthetic have been pro-posed [32,33]. As reported earlier, it has been proved that fungi have an effective enzymatic repertoire to alleviate the reactive nitrogen species (such as peroxynitrite (ONOO−) and nitric oxide (NO)) produced by the host [33]. Previous published literature has found that the performance of the fungus Neurospora сrassau was affected by changing the cultivation medium with nitric oxide donor (sodium nitroprusside, SNP or S-nitrosoglutathione) and NOS inhibitor (L-nitroarginine) [34]. Moreover, Wang et al have demonstrated that fungal secondary metabolite biosynthesis would be regulated and generated the production of photoactive perylenequinones by SNP [35]. Therefore, based on the reports aforementioned, considering the imbalance of NO content in neuroinflammation, we proposed a strategy to force fungus to alter the overdose of NO condition, then generate some chemical agents for treating neurodegeneration-related diseases.

For the comment: “The authors ignored the recommendation to discuss the anti-inflammatory activity of the compounds”, in the related edition, we have added this parts, as follows: Moreover, we also analyzed the anti-inflammation activities of those differential metab-olites in Figure S1-S4. We have found that compound 27 (Agnuside) in Figure S1 could be a nutraceutical for allergic asthma [36]. Similarly, eucalyptol (12, in Figure S2) could reduce the expression level of inflammation cytokines, and control airway mucus hyper-secretion [37]. Corticosterone (11, in Figure S3) has been reported as a treatment for alcoholic tissue injury at the Gut-Liver-Brain axis, acting on alcohol-induced expression of inflammatory cytokines [38]. For compound corticosterone isorhamnetin-3-glucoside (4, in Figure S4), has been regarded as the principal substance responsible for anti-inflammatory activity [39].

For the comment “In the discussion (lines 396-409), the results are essentially repeated and there is no analysis of the literature on already known literary data. I.e., the authors need to discuss what activity has already been shown for substances, and how this can ensure their activity described in this manuscript”. we have deleted those content, and we also re-analyzed the activities of SF1-SF9 based on the reported literature, as follows: Furthermore, we measured the anti-neuroinflammation activities of these extracts and isolates (SF1-SF9) from AS1, in LPS-induced BV2 cells. Among them, compound SF4 is the same one identified from the extract of AS1 (Figure S1, compound 18), and SF3-SF5 all belong to the family of indole alkaloids deriving from the tryptophan, compound SF8 was isolated from the genus of Aspergillus for the first time. In addition, SF4 (in-dole-3-carboxaldehyde) could inhibit intestinal inflammation on the TLR4/NF-kB/p38 signaling pathway, and balance amino acid metabolism in mice with colitis [40]. 2-Acetamidophenol (SF6) is an aromatic compound with the potential for anti-inflammatory activity, and has been widely used in the pharmaceutical industry [41]. SF8 (terphenyllin) could suppress growth and metastasis of gastric cancer by inhibiting the STAT3 signaling pathway [42]. Taken together, our results indicated that alkaloids and nitrogen-containing compounds generated by Aspergillus after SNP was administered have shown a potential to treat inflammation-related disease, while in-depth biological testing needs further investigation.

  1. The remark was taken into account.

Response: Thank you.

  1. In the Discussion, the authors took into account the remark that the additives used were probably only an additional source of nitrogen for the biosynthesis of nitrogen-containing compounds. However, nothing has changed in setting the goal. That is, now the discussion of the results does not correspond to the goal. Or the goal does not match the results. In general, you need to revise the entire manuscript from beginning to end.

Response: thank you for your comments, we have revised the goal, and re-changed the discussion part.

  1. The authors identified several compounds and described the isolation scheme. However, there is no information about how these compounds were identified (their structure was established). Are these substances new or known?

Response: thank you for your comments, all of those compounds are known, and as you suggest, their LCMS and NMR data were provided in Figure S5-S30, and Table S1-S5.

  1. The headings of some tables and figure captions are incorrect.

Response: Thank you for your consideration, we have revised the headings of figure 3, table 1, figure 4, figure 6.

Round 2

Reviewer 1 Report (New Reviewer)

The authors answered my questions, except for one:

Fig. 1 (C) - how can we explain the fact that when 0.3 mM NO synthetase inhibitor was added to the tested samples, a significant decrease in NO accumulation was observed, while increasing or decreasing the concentration of L-NAME reduced these values by more than half, the same is true for the experiment with the addition of sodium nitroprusside.

I would like the authors to explain this strange "dose-dependent effect"

The authors answered my questions, except for one:

Fig. 1 (C) - how can we explain the fact that when 0.3 mM NO synthetase inhibitor was added to the tested samples, a significant decrease in NO accumulation was observed, while increasing or decreasing the concentration of L-NAME reduced these values by more than half, the same is true for the experiment with the addition of sodium nitroprusside.

I would like the authors to explain this strange "dose-dependent effect"

Author Response

Fig. 1 (C) - how can we explain the fact that when 0.3 mM NO synthetase inhibitor was added to the tested samples, a significant decrease in NO accumulation was observed, while increasing or decreasing the concentration of L-NAME reduced these values by more than half, the same is true for the experiment with the addition of sodium nitroprusside.

I would like the authors to explain this strange "dose-dependent effect"

Response: Thank you for your consideration. In our previous study, we found that  under the differential concentration of SNP/L-NAME administrated, the yield of extract was different, and the peak areas of differential metabolites on HPLC profiles were also distinctive, therefore, maybe an appropriate concentration of exogenous nitric oxide will act on the expression of differential metabolites, which are responsible for anti-inflammation activity. And this strange phenomenon of "dose-dependent effect" is in accordance with the literatures as Wang et al (Figures S3-S4) [1], Min et al (Figure 1)[2], Chen et al (Figures 1-2) [3], Mahendran et al (Figures 3-5) [4], and Li et al (Figure 1)[5] reported.

  1. Wang, W.J.; Li, X.P.; Shen, W.H.; Huang, Q.Y.; Cong, R.P.; Zheng, L.P.; Wang, J.W. Nitric Oxide Mediates Red Light-Induced Perylenequinone Production in Shiraia Mycelium Culture. Bioresources and Bioprocessing 2024, 11, 2, doi:10.1186/s40643-023-00725-5.
  2. Min, W. F.; Fang, J. Y,; Shi, Y. F.; Bai, X. R.; She, Y. . F.; Tian, H. T.; Luo, C. K. Effect of Sodium Nitroprusside Addition on Carbon and Nitrogen Metabolism of Rice during Germination under Alkali Stress. Journal of Plant Nutrition and Fertilizers 2024, 30, 934–947.
  3. Chen, S. H.; Hang, Z. Y.; Zhang, H. C.; Liu, Y.; Yang, W. X. The Effect of Exogenous Sodium Nitroprusside on Contents of Active Substances and In-Vitro Antioxidant Capacity of Cyclocarya Paliurus Leaves. Shandong Forestry Science and Technology 2024, 54, 1–7.
  4. Mahendran, G.; Kumar, D.; Verma, S.K.; Chandran, A.; Warsi, Z.I.; Husain, Z.; Afroz, S.; Rout, P.K.; Rahman, L.U. Sodium Nitroprusside Enhances Biomass and Gymnemic Acids Production in Cell Suspension of Gymnema Sylvestre (Retz.) R.Br. Ex. Sm. Plant Cell Tiss Organ Cult 2021, 146, 161–170, doi:10.1007/s11240-021-02058-7.
  5. Li, M.; Peebles, C.A.M.; Shanks, J.V.; San, K.-Y. Effect of Sodium Nitroprusside on Growth and Terpenoid Indole Alkaloid Production in Catharanthus Roseus Hairy Root Cultures. Biotechnology Progress 2011, 27, 625–630, doi:10.1002/btpr.605.

Reviewer 2 Report (New Reviewer)

The authors need to correct a number of errors

The authors provided references to the works [1-4] in their response to my comments, but they were not added to the text. Obviously, these references and explanations should be added to the Introduction (roughly lines 64-65).

 In Discussion, the statement «compound SF8 413 was isolated from the genus of Aspergillus for the first time» (Line 413-414) does not correspond to reality because earlier terphenyllin was repeatedly isolated from Aspergillus candidus [https://www.jstage.jst.go.jp/article/antibiotics1968/28/4/28_4_328/_article/-char/ja/] and [https://link.springer.com/article/10.1007/s10600-017-2108-y], Aspergillus taichungensis [https://pubs.acs.org/doi/abs/10.1021/np2000478] as well as another Aspergillus fungi. Moreover, neuroprotective effect of terphenyllin was studied in [https://www.mdpi.com/1420-3049/26/12/3618].

Author Response

Review2:

The authors provided references to the works [1-4] in their response to my comments, but they were not added to the text. Obviously, these references and explanations should be added to the Introduction (roughly lines 64-65).

Response: Thank you for your consideration. We have added it to the text and marked it in red (in the newest version, lines 65-68).

In Discussion, the statement “compound SF8 413 was isolated from the genus of Aspergillus for the first time» (Line 413-414) does not correspond to reality because earlier terphenyllin was repeatedly isolated from Aspergillus candidus [https://www.jstage.jst.go.jp/article/antibiotics1968/28/4/28_4_328/_article/-char/ja/] and [https://link.springer.com/article/10.1007/s10600-017-2108-y], Aspergillus taichungensis [https://pubs.acs.org/doi/abs/10.1021/np2000478] as well as another Aspergillus fungi. Moreover, neuroprotective effect of terphenyllin was studied in [https://www.mdpi.com/1420-3049/26/12/3618].

Response: Thank you for your consideration. We have deleted the sentence “compound SF8 413 was isolated from the genus of Aspergillus for the first time” in our revised edition. And added the sentence: also showing significant neuroprotective effects against oxidative stress in Neuro-2a Cells[46] in line 421.

Round 3

Reviewer 1 Report (New Reviewer)

The authors responded to all my comments.

The authors responded to all my comments.

This manuscript is a resubmission of an earlier submission. The following is a list of the peer review reports and author responses from that submission.

Round 1

Reviewer 1 Report

In this research the authors used sodium nitroprusside and L-NG-nitroarginine methyl ester to modify of the metabolism of Aspergillus sp. fungus. Moreover, the effect of the extracts on LPS-stimulated microglia cells were aimed.

The work looks quite interesting, but in its current form there are a lot of questions about it. Therefore, it should not be accepted for publication without serious revision.

Thus,

1. The fungus was identified using ITS sequencing, but authors did not present the phylogenetic tree. However, the section 3.1. was named Phylogenetic analysis. Unfortunately, it is obvious that no phylogenetic analysis is presented in the manuscript. Instead, scant statistics on publications on Aspergillus fungi are provided. The meaning of these article statistics is unclear, since the authors do not show a phylogenetic tree based on the results of a molecular genetic study and it is unclear which species their fungus is close to. The authors should show a phylogenetic tree indicating the maximum proximity to a particular strain, so that it can be considered that the fungus has actually been identified. Figure 1A should be deleted.

2. The name of the section 3.2. “Fermentation conditions of secondary metabolites produced by SYPUF29 with or without NO stress” has no any sense. The name should be corrected.

3. The purpose of the study must be absolutely clearly formulated. The authors write: ”In this study, we planned to obtain more novel neuroinflammatory inhibitors from 62 the secondary metabolites of A. sp. SYPUF29under NO stress.”  However, there is no isolation of new individual compounds in the work. The authors conducted a study of the effect of additives on the production of secondary metabolites and evaluated the anti-inflammatory activity of extracts. If this is their goal, then it needs to be formulated that way.

4. Based on the above, the author needs to change the sequence of description of the Results. First, the effect of cultivation conditions on the metabolic profile, then the biological activity of the extracts.

5. The authors write: “Based on the exact masses, fragmentation pathways, retention behaviors and related botanical biogenesis, botanical biogenesis, these compounds were identified.” The authors conducted a large statistical analysis of the HPLC-MS data obtained, but did not provide any data on the identification of compounds. There is no detailed description of how the compounds were identified. This must be added either to the Appendix or to Supporting Files. Currently, a large number of works in this field have been published so that the authors can take them as a sample.

6. The authors are very careless with the text. What is SYPUF29 and F29?

7. The using of sodium nitroprusside as an exogenous NO donor and L-NAME as a NO synthetase inhibitor should be justified by links to relevant publications in the Introduction. Moreover, cultivation with isosorbide dinitrate (ISO) suddenly "appeared" in the section. But nothing is said about this in the Introduction. The authors need to describe in all sections of the manuscript what they added to the cultural environment and why.

8. Figure 1c is completely unclear and needs to be corrected. It is titled “Anti-inflammatory activities of the metabolites under different concentrations of SNP and L-NAME on LPS-induced NO production in BV2 microglial cells.” but this is a misnomer. The table should be designed according to the rules of the journal and it should be clearly indicated which parameter is reflected in it. What is the IC50 in this table?

9. Figure 2 is titled “The effect of the metabolites on the expressions…..”, but this is a error. The authors did not investigate the activity of metabolites. The activity of the extract was investigated. Therefore, the signatures must be adjusted. In addition, the Methods section does not describe how the extracts were prepared for biological experiments, in which solvent they were introduced, and in what concentration. There is only a description of the preparation of ethyl acetate extract for HPLC-MS.

10. Table S5 is absence in the manuscript.

11. The discussion requires significant refinement. The authors should discuss several aspects. 1. What biological activity is known for up-regulated compounds? Has anti-inflammatory activity been described for them before? 2. Which secondary metabolites were previously actually isolated from this fungus? 3. Does the appearance of new compounds correspond to the available information about the reaction of fungi to NO stress?

12. The authors, like many others, write about dormant genes that can be awakened by OSMAC strategy. Where is the evidence that in this case some new genes are waking up, and there is no shift in enzyme activity, for example?

13. Only nitrogen-containing compounds were up-regulated after sodium nitroprusside and L-NG-nitroarginine methyl ester were used. Both of these compounds are nitrogen-containing. Thus, the fungus could use them as a source of nitrogen for its metabolism. There are numerous studies that show that the addition of nitrogen-containing compounds to the culture medium leads to the formation of nitrogen-containing metabolites. It's so logical. In this regard, the authors need to prove that these compounds could not have been metabolized by the fungus if the authors want to maintain their stress statement. Or, the authors need to reconsider their concept and recognize that an additional source of nitrogen allowed the fungus to produce large amounts of compounds. This does not negate the anti-inflammatory effect of the extracts and the value of the work will not become less. But conclusions must be drawn correctly and reasonably.

Reviewer 2 Report

Xiao et al. changed the metabolic profile of Aspergillus sp. SYPUF29 by adding NO donor or NO synthetase inhibitor and found that the extracts of SYPUF29 treated with SNP or L-NAME showed more potent anti-inflammatory activity than those of untreated fungi. The anti-inflammatory activities of the extracts were confirmed by ELISA, real-time PCR, and Western Blotting. To identify compounds showing anti-inflammatory activity, the authors compared the metabolites of the three extracts and found the amounts of several metabolites changed among the three. However, the authors made critical mistakes in the identification of the compounds. They used mzCloud, mzVault and MassList databases where few fungal secondary metabolites are listed. Using such datasets, primary metabolites common among most living organisms can be identified but secondary metabolites cannot. For this reason, 44 key compounds shown in Figure 6 seem to be misassigned; most of them are not natural products but synthetic substances. As the identification of the compounds is incorrect, the discussion based on the compounds makes no sense. 

The reviewer suggests 2 plans to the authors to improve this manuscript. One plan is to identify the species of SYPUF29 and list secondary metabolites reported to be produced by closely related species. Then, predict the structure of metabolites. The other plan is to isolate active compounds and elucidate the structures of them.

In lines 68 and 77, Aspergillus sp. cannot be abbreviated as “A. sp.”. Furthermore, “sp.” should be plain, not italic.

In section “2.5 Determination of Cell Viability”, how was formazan dissolved? Insoluble formazan should be formed after the addition of MTT.

In section “3.1 Phylogenetic analysis”, the authors only mention the distribution of species of Aspergillus. Which section does SYPUF29 belong to? The authors should conduct phylogenetic analysis based on the sequences of ITS region, β-tubulin, and RNA-polymerase.

In Figure 1c, the table is not reader-friendly. It appears that L-NAME-0.1mM exhibited IC50 57.29 µg/mL although the value is responsible for the extract of SYPUF29 treated with L-NAME-0.1mM. Moreover, there is no explanation of MINO.

In Figure 5 and Figure 7, as mentioned in major comments, the identification of metabolites went wrong. Classification of the compounds should be done after the identification is conducted correctly.

In Figure 6A, there are many synthetic compounds included. For example, compound 20 is 7-aminonimetazepam, a metabolite of a hypnotic drug nimetazepam. Identification of the compounds should be conducted correctly.